# SAGE: A Synchronized Action and Gaze Estimation Framework for Comprehensive Human Behavior Analysis

## Abstract

Human object interactions, gaze patterns, and their anticipation are intricately linked, providing valuable insights into cognitive processes, intentions, and behavior. However, most existing models handle gaze and actions separately, missing both their interdependence and the advantages of a unified solution. This paper presents a novel unified framework, SAGE (**S**ynchronized **A**ction and **G**aze **E**stimation), which integrates simultaneous recognition and anticipation of both human object interaction and human gaze into a single unified end-to-end trainable model. Our approach leverages a transformer-based architecture and incorporates gaze data into spatiotemporal attention mechanisms to simultaneously predict current and future human actions and gaze behavior. We explore this bidirectional relationship between gaze and actions under different scenarios, whether requiring a close-up, detailed view (egocentric) or a wider, more contextual view (exocentric), making our framework versatile for various applications. Additionally, due to lack of datasets for comprehensive analysis of both human object interactions and gaze in exocentric videos, we establish a new benchmark *Exo-Cook* to facilitate further research in this domain. Experiments on three benchmark datasets—VidHOI, EGTEA Gaze+, and Exo-Cook—show that jointly modeling gaze and actions across current and future frames achieves consistently strong results, often surpassing specialized state-of-the-art models tailored to individual tasks. By unifying actions and attention in a comprehensive way, our work lays the groundwork for more intuitive human-machine interaction and future applications in cognitive rehabilitation and behavior analysis.

## 1 Introduction

Imagine watching a person in a kitchen. They glance at a knife, reach for a cutting board, and then shift their gaze toward a tomato. Even before they pick up the knife, you can reasonably anticipate what comes next: they're about to start chopping. This ability to recognize current actions and predict future ones based on where someone is looking and how they interact with objects is something humans do effortlessly—and crucially, through a single, unified cognitive model in the brain, not through fragmented or disjointed processes. For intelligent systems to interact naturally with people, they must unify perception and prediction—understanding both current and future behavior without relying on separate task-specific pipelines. From assistive robots to driver monitoring and AR assistants, many applications require not only understanding current actions but also anticipating what comes next. Gaze offers cues of intention, while human–object interactions reflect engagement with the environment. Crucially, the combination of these two modalities, and how they unfold over time offers the richest behavioral insight.

Yet, most existing approaches treat these tasks separately: some recognize human-object interactions and actions Ji et al. (2021); Cong et al. (2021); Tu et al. (2022); Chiou et al. (2021); Mascaro et al. (2023); Ni et al. (2023); Hao et al. (2022); Wang et al. (2020) or anticipate them Roy et al. (2024); Girdhar & Grauman (2021a); Zhong et al. (2023); Liu et al. (2020); Cong et al. (2021); Ni et al. (2023), while others model gaze prediction Lai et al. (2024b); Li et al. (2021); Lai et al. (2024a); Huang et al. (2018); Tafasca et al. (2024); Chong et al. (2020) or gaze anticipation Lai et al. (2024c); Zhang et al. (2017). Few attempt to integrate both, and those that do either use disjointed pipelines

or focus only on egocentric views Li et al. (2015); Ma et al. (2016); Singh et al. (2016); Ni et al. (2023); Li et al. (2018a; 2021); Min & Corso (2021). Importantly, most do not model how gaze and actions evolve together into the future—a key aspect of joint anticipation and intuitive, human-like understanding.

We propose **SAGE** (Synchronized Action and Gaze Estimation), a unified end-to-end framework that jointly recognizes and anticipates gaze and human actions. The "Estimation" for gaze and action refers to both present and future. Built on a transformer backbone, SAGE integrates gaze into spatiotemporal attention to learn shared representations that capture the coupling between where people look and what they do. Central to our design are two modules: the Gaze-Conditioned Spatial Attention (GCSA) module, which injects gaze into spatial attention to highlight human–object interaction cues, and the Gaze-Conditioned Temporal Prediction (GCTP) module, which models temporal correlations between future gaze and actions. Unlike prior methods limited to either egocentric or exocentric data, our modular framework supports both through separate training, enabling consistent reasoning and broader applicability—without relying on fragmented, task-specific pipelines.

However, training and evaluating such models demands appropriate datasets. Most existing datasets focus exclusively on either actions Damen et al. (2018); Kuehne et al. (2014); Lea et al. (2016) or gaze Chong et al. (2020); Tafasca et al. (2023), and the few that include both are primarily limited to first-person perspectives Li et al. (2021); Grauman et al. (2022). While Ego-Exo4D Grauman et al. (2024) provides third-person annotations for both gaze and actions, it lacks a benchmark for joint modeling and evaluation. Moreover, Ego-Exo4D is not directly usable for this task—it requires preprocessing of annotations, new label creation, modality alignment, and task-specific structuring for joint gaze–action modeling. To address this, we introduce Exo-Cook, a third-person benchmark derived from Ego-Exo4D, specially curated for evaluating unified models of gaze and action recognition and anticipation. Exo-Cook fills a critical gap and enables research on predictive human behavior in third-person settings.

In summary, our contributions are four fold. **First**, we propose a unified end-to-end trainable architecture that integrates recognition and anticipation of both human-object interaction (HOI) and gaze, allowing for joint optimization of these tasks for comprehensive human behavior understanding. **Second**, we introduce a Gaze Conditioned Spatial Attention (GCSA) module that provides human-object interaction cues in the spatial domain and a Gaze Conditioned Temporal Prediction (GCTP) module which simultaneously models temporal correlations between future gaze patterns and future actions. **Third**, unlike prior works that focus on either egocentric or exocentric views, our modular framework supports both through separate training, enabling broader applicability across diverse scenarios. **Fourth**, we demonstrate the efficacy of our framework against state-of-the-art methods and introduce Exo-Cook, a new benchmark for evaluating models that integrate human-object interaction and gaze analysis in third-person videos—providing a comprehensive foundation for future research in this area.

## 2 RELATED WORKS

### 2.1 GAZE DETECTION AND ANTICIPATION

Gaze detection focuses on estimating a person's current visual attention point and has been studied in both egocentric and exocentric settings. The GazeFollow dataset Recasens et al. (2015) laid the foundation for static gaze-following in images and was later extended to videos Recasens et al. (2017); Chong et al. (2020); Tafasca et al. (2024). Early models typically used two-stream architectures that fused head pose and saliency cues Chong et al. (2018); Horanyi et al. (2023), while recent methods have enhanced gaze localization by incorporating depth information Tafasca et al. (2023); Bao et al. (2022); Hu et al. (2023) and human pose Gupta et al. (2022). While the primary goal of these models is accurate gaze localization, gaze is often used downstream to aid action recognition. In contrast, leveraging action to inform gaze has been rarely explored and may even hurt the performance Li et al. (2018b; 2021). In this work, we propose a joint estimation framework to demonstrate that action understanding and gaze detection enhance each other in a complementary manner.

Gaze anticipation, in contrast, focuses on forecasting where gaze will shift in the future. This task has been studied *exclusively* in egocentric settings, where gaze shifts often precede interactions with objects. Existing approaches Lai et al. (2024c); Zhang et al. (2017) model gaze trajectories over time by leveraging motion cues and temporal scene context. However, these methods treat gaze

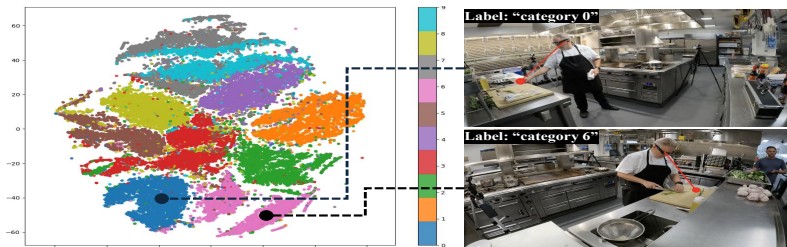

Figure 1: **Left**: Exo-Cook labels clustered and projected using t-SNE. **Right**: Visualization examples from two clusters (category 0 and 6) with atomic action descriptions and their verb–noun annotations - (Top) *C picks a strainer from the countertop with his right hand"* [Verb: picks; Noun: strainer]. (Bottom) *"C cuts the green onion with a knife on the cutting board"* [Verb: cuts; Nouns: green onion, knife, cutting board].

anticipation as an isolated task and do not account for its interplay with actions—limiting their understanding of intent. More importantly, to our knowledge, no existing work has explored gaze anticipation in exocentric videos—leaving a critical gap in third-person behavior modeling.

## 2.2 RECOGNIZING AND ANTICIPATING ACTIONS

Gaze has proven valuable in enhancing action recognition by guiding attention to informative regions. Early approaches Wang et al. (2016); Chen et al. (2014); Mathe & Sminchisescu (2012); Shapovalova et al. (2013) used saliency or gaze cues alongside handcrafted features, while later works employed learned attention without relying on real gaze data Sharma et al. (2016). More recent models have integrated gaze more explicitly: Li et al.Li et al. (2021) jointly model gaze and action, using predicted gaze to guide feature selection, and Ni et al.Ni et al. (2023) incorporate gaze heatmaps into a multimodal Transformer for HOI detection. These approaches demonstrate that gaze improves spatial localization and contextual precision in action recognition.

Meanwhile, action anticipation aims to forecast future behavior from partial observations. Existing methods often model HOIs using spatio-temporal graphs Jain et al. (2016); Materzynska et al. (2020); Ou et al. (2022); Teng et al. (2021); Wang & Gupta (2018), supported by rich datasets Girdhar & Grauman (2021c); Damen et al. (2022); Teed & Deng (2020) and architectural advances like RU-LSTM Furnari et al. (2019), AVT Girdhar & Grauman (2021b), MemViT Fu et al. (2022), RAFTformer Girase et al. (2023) and InAViT Roy et al. (2024). Recent trends explore goal-conditioned reasoning Roy & Fernando (2022b;a), motion primitives Dessalene et al. (2023), and multimodal cues such as audio Wu et al. (2021). Yet, despite its predictive power, gaze remains underutilized in action anticipation. Very few approaches explicitly model the temporal link between future gaze and future actions, leaving a significant gap in understanding how attention and interaction co-evolve over time.

Together, these observations point to the need for unified models that go beyond static perception, capturing the bidirectional and predictive relationship between gaze and actions—both in the present and the future.

## 3 EXOCOOK DATASET

The Ego-Exo4D dataset Grauman et al. (2024) is a large-scale human activity dataset captured from both egocentric and exocentric views, spanning 8 domains, 740 participants, and 123 scenes. It provides time-aligned descriptions and dense annotations such as 3D pose, gaze, and object masks, though these are not directly usable for deep model training. For our study, we use the exocentric videos from the "Cooking" domain, which naturally suits HOI–gaze analysis, and construct Exo-Cook through extensive label generation and task-specific adaptation. Specifically, we extract 658 cooking videos and 189,225 textual descriptions from Ego-Exo4D and preprocess them as follows: **(a) Human bounding boxes:** using the pipeline in Ni et al. (2023) with YOLOv5 Jocher et al. (2022) to detect full-body and head boxes of the camera wearer in third-person views. **(b) Object bounding boxes:** generating boxes for interactable objects from instance masks provided in the Ego-Exo4D metadata. **(c) Gaze heatmaps:** We go through four steps to generating gaze heatmaps.

Table 1: **Exo-Cook data distribution**: Row A shows sample counts per category after clustering; Row B shows counts after removing invalid samples.

| | 0 | 1 | 2 | 3 | 4 | 5 | 6 | 7 | 8 | 9 | Total |
|---|---|---|---|---|---|---|---|---|---|---|---|
| A. Clustering | 2562 | 2741 | 1210 | 688 | 2675 | 4319 | 4122 | 3530 | 5812 | 5662 | 33321 |
| B. Valid Samples | 2562 | 2741 | 1193 | 688 | 2675 | 4167 | 4000 | 3530 | 5500 | 4994 | 32050 |

First, we calculate 3D gaze intersection point $G_{3D}^{aria}$ from binocular gaze vectors. Second, transform $G_{3D}^{aria}$ to the third-person camera space via the relative pose, yielding $G_{3D}^{camera}$. Third, project $G_{3D}^{camera}$ to 2D as $G_{2D}^{camera}$. Fourth, generate a 2D Gaussian heatmap $M^{\text{pseudo}}$, centered at $G_{2D}^{camera}$, with standard deviation: $\sigma = \frac{W_{\text{hm}}+H_{\text{hm}}}{2} \cdot \frac{3}{64}$ where $(W_{\text{hm}}, H_{\text{hm}})$ denotes the heatmap size. **(d) Action labels:** We apply spaCy Explosion (2024) to extract all possible verbs and nouns from the textual action descriptions and then apply BERT Devlin et al. (2019) to generate semantic embeddings of the simplified annotations. We use K-means clustering ($K = 10$), chosen via the Elbow Method, to group semantically similar actions into 10 categories indexed 0–9. We show the clusters in Figure 1. **(e) Label alignment:** These labels are aligned with video timestamps, and we use DeepSORT Wojke et al. (2017) to track human–object trajectories. Exo-Cook contains 33,321 video clips with atomic action descriptions, from which we remove samples lacking gaze labels, valid head boxes, or proper alignment. This yields 32,050 valid clips annotated with bounding box, gaze, and action labels. The distribution is shown in Table 1. We split the data into 25,650 training, 3,200 validation, and 3,200 test samples. Further details are provided in the Appendix (Section B).

# 4  SAGE

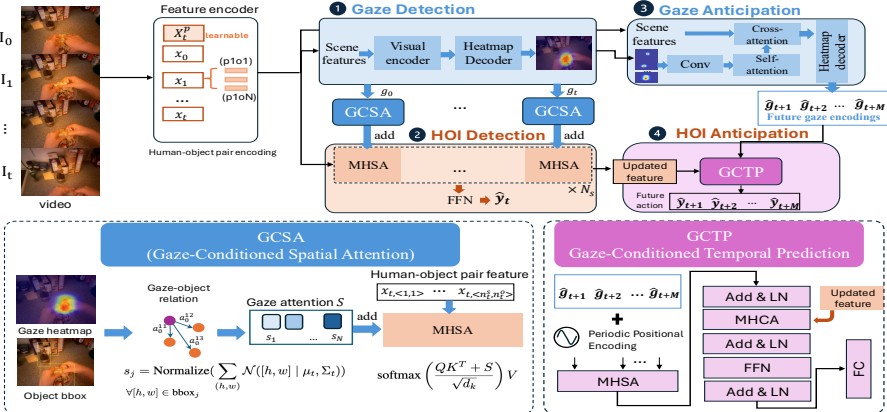

Figure 2: **Overview of SAGE.** The framework unifies gaze detection, human–object interaction (HOI) detection, and anticipation tasks. The Feature encoder extracts appearance and object-location features as human–object pairs. The *Gaze Detection* module predicts a probability map of the fixation point, which is integrated into *HOI detection* via the proposed **GCSA** module. A *Gaze Anticipation* module predicts future gaze from observed images and gaze features, while the **GCTP** module enables *HOI Anticipation* conditioned on action and future gaze features.

In this section, we present the SAGE framework and describe its training objective, as illustrated in Figure 2. Let us consider the anticipation task of predicting activity labels $\boldsymbol{y}$ given a short video clip $X_t = I_{0:t}$. We want to predict future activity at $t + M$ with $M$ steps, i.e. $\boldsymbol{y}_{t:t+M} = [y_t, y_{t+1}, \ldots, y_{t+M}]$, conditioned on future gaze, i.e. $\boldsymbol{g}_{t:t+M} = [g_t, g_{t+1}, \ldots, g_{t+M}]$. Therefore, we decompose the problem as: $p(y_{t:t+M} \mid I_{0:t}) = \int_G p(\boldsymbol{y}_{t:t+M}, \boldsymbol{g}_{0:t+M} \mid I_{0:t})\, dG = \int_G p(\boldsymbol{y}_{t:t+M} \mid \boldsymbol{g}_{0:t+M}, I_{0:t})\, p(\boldsymbol{g}_{0:t+M} \mid I_{0:t})\, dG = \int_G p(y_t \mid \boldsymbol{g}_{0:t}, I_{0:t})\, p(\boldsymbol{y}_{t+1:t+M} \mid y_t, \boldsymbol{g}_{t+1:t+M}, I_{0:t}) \cdot p(\boldsymbol{g}_{0:t} \mid I_{0:t})\, p(\boldsymbol{g}_{t+1:t+M} \mid \boldsymbol{g}_{0:t}, I_{0:t})\, dG$, where $p(\boldsymbol{g}_{0:t} \mid I_{0:t})$ is the *gaze detection* module (Module 1) from video appearance, $p(y_t \mid \boldsymbol{g}_{0:t}, I_{0:t})$ is the *HOI detection* module (Module 2) conditioned on gaze and image appearance, $p(\boldsymbol{g}_{t+1:t+M} \mid \boldsymbol{g}_{0:t}, I_{0:t})$ is the *gaze anticipation* module (Module 3) predicting future gaze for $M$ steps, and $p(\boldsymbol{y}_{t+1:t+M} \mid y_t, \boldsymbol{g}_{t+1:t+M}, I_{0:t})$ (Module 4) performs *HOI anticipation* conditioned on video features, predicted gaze, and current action.

Note that our framework supports both HOI and standard action labels, and we use these terms interchangeably throughout the paper as they both denote interactions. Egocentric datasets (e.g., EGTEA) use the term "action" since the human is not visible, while exocentric datasets use "HOI" due to visible human presence—but both represent the same concept in our setting.

### 4.1 GAZE DETECTION

As illustrated in Figure 2, our gaze detection module consists of a visual encoder and a decoder that outputs gaze fixation heatmaps. The input video is divided into non-overlapping patches, flattened, and projected into a $D$-dimensional embedding space via linear mapping. These tokens are processed by transformer layers with self-attention. To produce gaze heatmaps, a transformer decoder—built on multiscale self-attention from MViT Fan et al. (2021)—upsamples the encoded features into maps of size $T'H'W' \times D'$, followed by a softmax to yield the final heatmap $g_{0:t}$. For egocentric videos, we adopt GLC Lai et al. (2024a), while for exocentric videos we use the architecture of Chong et al. (2020), initializing with their pretrained weights.

**Gaze-conditioned Spatial Attention (GCSA).** Our HOI recognition model predicts $p(y_t \mid g_{0:t}, I_{0:t})$, current action $y_t$ is predicted conditioned the gaze $g_{0:t}$ and video feature $I_{0:t}$. Given the object bounding box for object $j$ in the image, we compute a gaze-conditioned score $s_{t,j}$. Denoting the predicted gaze heatmap from module 1 as gaze distribution $p(g_t) \sim \mathcal{N}([\mu_t, \Sigma_t])$, $s_{t,j}$ is calculated as

$$s_{t,j} = \text{Normalize}(\sum_{(h,w)} \mathcal{N}([h,w] \mid \mu_t, \Sigma_t)) \tag{1}$$

where $[h, w]$ is sampled within the $\text{bbox}_j$. We compute $s_j$ for each human-object pair and generate the Gaze-conditioned score matrix $S_t$ for every frame, i.e., $S_t = [s_{t,j=1}, ..., s_{t,j=N}]$. We apply $S$ as an attention bias in the Multi-Head Self-Attention layer of transformer and we name it as Gaze-conditioned Spatial Attention (GCSA).

$$\text{GCSA}(Q, K, V, S) = \text{softmax}\left(\frac{QK^T + S}{\sqrt{d_k}}\right) V \tag{2}$$

where $S \in \mathbb{N}^{B \times N \times N}$, $B$ is batch size, $N$ is the largest number of pairs in each sequence.

### 4.2 HOI DETECTION

Inspired by Ni et al. (2023), we adopt a spatio-temporal transformer architecture for HOI detection, consisting of a spatial encoder and a temporal encoder. The spatial encoder exploits human-object appearance representations from each frame to understand spatial relations between human and all the possible objects in the scene. The spatial encoder receives the human-object representations $X_t = [\mathbf{x}_{t,\langle 1,1\rangle}, \ldots, \mathbf{x}_{t,\langle i,j\rangle}, \ldots, \mathbf{x}_{t,\langle n_t^s, n_t^o\rangle}]$ within one frame as the input. For egocentric videos, $n_t^s = 1$ and $n_t^0 o$ is the number of detected objects in frame $t$. One major difference with Ni et al. (2023) is that we attach one learnable global token $X_t^p$, representing the global representation of frame $t$. We exploit stacked $X_t^p$ with $X_t$ as input to the HOI detection model. After $N_d$ stacked Transformer self-attention layers (MHSA), the global token summarizes the dependencies between human-object pairs to the global appearance feature vector, while the pair relation representations are refined to $\mathbf{X}_t^{sp}$.

The GCSA module introduced in last section is integrated into the MHSA layer. The GCSA-enhanced self-attention is defined as: $\boldsymbol{q}_t^l = \text{LN}(\text{GCSA}(Q_t^{(l-1)}, K_t^{(l-1)}, V_t^{(l-1)}, S_t) + \boldsymbol{q}_t^{(l-1)}), l = 1, \ldots, N_d$, $N_d$ is the layer number. Then we apply FFN (feed forward network) to $\boldsymbol{q}_t^l$: $\boldsymbol{c}_t = \text{FFN}(\boldsymbol{q}_t^{N_d})$ The token $\boldsymbol{c}_t$ encodes various spatial feature, including gaze-to-object correlation and human-object spatial relations. To model temporal relation through the input sequence, we use cross-attention over frame-level global features. We first add a Periodic Positional Encoding (PPE) as proposed by FaceFormer Fan et al. (2022) to the token $\boldsymbol{c}_{0:t}$, denoted as $\hat{\boldsymbol{c}}_k = \boldsymbol{c}_k + \text{PPE}(k), k = 0, \cdots, t$. Then we apply MHSA layers to implicitly encode the temporal correlation of actions, denoted as:

$$\hat{\boldsymbol{c}}_t^l = \text{LN}(\text{MHSA}(Q^{(l-1)}, K^{(l-1)}, V^{(l-1)}) + \hat{\boldsymbol{c}}_t^{(l-1)}), l = 1, \cdots, N_d \tag{3}$$

Then our model predicts the action for the last frame in the input sequence through a FFN layer and a MLP layer, denoted as $\hat{y}_t = \text{MLP}(\text{FFN}(\{\hat{\boldsymbol{c}}_t^{(N_d)}\}))$.

### 4.3 Gaze Anticipation

The goal of the gaze anticipation model is to predict the future gaze $\boldsymbol{g}_{t+1:t+M}$ based on current gaze $G_{0:t}$, input video $I_{0:t}$, and the previous gaze encoder output. We design a Transformer-based model to anticipate future gaze. With the predicted gaze heatmap sequence $\boldsymbol{g}_{0:t}$ from the gaze detection model, we apply convolution layers to generate gaze feature vector. The gaze feature vectors are added to PPE and then feed to temporal layer as $\hat{\boldsymbol{g}}_{0:t}$. The input to the gaze anticipation module is $\boldsymbol{g}_k = \text{Conv}(g_k) + \text{PPE}(k), \quad k = 0, \ldots, t$. The gaze anticipation model is described as $\hat{\boldsymbol{g}}_{t+1:t+M} = \text{Decoder}(\text{MHCA}(\hat{\boldsymbol{c}}_{0:t}), \text{MHSA}(\boldsymbol{g}_{0:t}))$. The MHSA($\cdot$) represents self-attention layers (similar to Eq. 3) to encode gaze temporal correlations among $\{\boldsymbol{g}_k\}_{k=0}^t$. The MHCA(Multi-Head Cross Attention) layer applies cross-attention among the past gaze feature and the video feature for anticipating future gaze. In layer $l$, the MHCA is formulated as $\hat{\boldsymbol{g}}_k^l = \text{LN}(\text{MHCA}(Q^{\hat{c}_t^{(l-1)}}, K^{\hat{g}_k^{(l-1)}}, V^{\hat{g}_k^{(l-1)}}) + \hat{\boldsymbol{g}}_k^{(l-1)})$.

### 4.4 HOI Anticipation

We propose the **Gaze Conditioned Temporal Prediction (GCTP)** module that models joint temporal dependencies among action $\hat{y}_t$ and gaze features $\{\hat{\boldsymbol{g}}_k\}_{k=t+1}^{t+M}$. The model is formulated as $\hat{y}_{t+M} = \text{GCTP}(\hat{\boldsymbol{g}}_{t+1:t+M}, \hat{\boldsymbol{c}}_t^{N_d})$. The GCTP module takes $M$ future gaze encodings $\hat{\boldsymbol{g}}_{t+1:t+M}$ and the refined representation $\hat{\boldsymbol{c}}_t^{N_d}$ in Eq. 3 as input. This module first apply self-attention layer that encode temporal correlation among the future gaze encoding $\hat{\boldsymbol{g}}_{t+1:t+M}$. Then cross-attention is applied among the gaze feature and action feature. The temporal relations among future actions and future gaze is implicitly learned. The tokens are initialized with gaze tokens: $\boldsymbol{a}_{(1)} = \hat{\boldsymbol{g}}_{t+1:t+M}$. Then the MHSA and MSCA layer are applied sequentially, formulated as: $\hat{\boldsymbol{a}}_1^l = \text{LN}(\text{MHSA}(Q^{\hat{a}^{l-1}}, K^{\hat{a}^{l-1}}, V^{\hat{a}^{l-1}}) + \hat{\boldsymbol{a}}^{l-1})$, $\hat{\boldsymbol{a}}_2^l = \text{LN}(\text{MHCA}(Q^{\hat{c}^{N_d}}, K^{\hat{a}^{l-1}}, V^{\hat{a}_{(1)}^{l-1}}) + \hat{\boldsymbol{a}}_1^{l-1})$. The future HOI anticipation is produced with a MLP layer: $\hat{y}_{t+1:t+M} = \text{MLP}(\text{FFN}(\hat{\boldsymbol{a}}_{(2)}^{N_d}))$.

### 4.5 Adaptability Across Viewpoints

Egocentric (FPV) and exocentric (TPV) scenarios have typically been studied separately within gaze and action understanding literature, as their feature characteristics differ substantially and are rarely addressed in a single model. To the best of our knowledge, SAGE is the first framework to generalize across FPV and TPV for four tasks, leveraging viewpoint-specific modalities in feature encoding and gaze detection while maintaining a consistent network flow and output format, as a unified framework. During the feature encoding phase, the primary distinctions manifest in the generation of human-object pairs $X_t = [\mathbf{x}_{t,\langle 1,1 \rangle}, \ldots, \mathbf{x}_{t,\langle i,j \rangle}, \ldots, \mathbf{x}_{t,\langle n_t^s, n_t^o \rangle}]$. For TPV videos, $\mathbf{x}_{t,\langle i,j \rangle}$ encodes the scene features, human body appearance, body location, and the spatial relationships between humans and objects, with the latter being especially important for recognizing human actions. The human index $i \geq 1$, since multiple active subjects may appear in a third-person scene. In contrast, for FPV videos, $\mathbf{x}_{t,\langle i,j \rangle}$ focuses on the spatial relationships between the hands and objects, as the global human location is not accessible from the egocentric viewpoint. In this case, the human index is fixed to $i \equiv 1$. Another revision between FPV and TPV arises in the gaze detection module. In TPV videos, SAGE estimate human gaze $g$ using head appearance and orientation features, following approaches such as Chong et al. (2020). In FPV videos, however, gaze is interpreted more as an attention region within the egocentric scene, and the gaze model relies on scene features to predict the gaze distribution.

### 4.6 Loss Function

**Heatmap Loss** $\mathcal{L}_{\text{hm}}$ is the visual attention heatmap loss, defined as the L2 loss between the predicted heatmap $g$ and the ground truth heatmap $g^{\text{gt}}$: $\mathcal{L}_{\text{hm},1} = \sum_{k=0}^{k=t} \|\hat{g}_k - g_k^{\text{gt}}\|_2^2$. We adopt similar loss for gaze anticipation model: $\mathcal{L}_{\text{hm},2} = \sum_{k=t+1}^{k=t+M} \|\hat{g}_k - g_k^{\text{gt}}\|_2^2$. **In-Out Loss** $\mathcal{L}_{\text{io}}$ is defined as the binary cross-entropy between the predicted in-out label $o_p$ and the ground truth label $o_{\text{gt}}$, indicating whether the gaze target is within the frame. This loss if only applied when the in-out label is available: $\mathcal{L}_{\text{io}} = \sum_t -o_t^{gt} \log(\hat{o}_t) - (1 - o_t^{gt}) \log(1 - \hat{o}_t)$. **Action Loss** $\mathcal{L}_{\text{act}}$ is used for action models. We apply Cross-Entropy loss for HOI detection, defined as $\mathcal{L}_{\text{act},1} = -y_t^{gt} \log \hat{y}_t$. For HOI anticipation model,

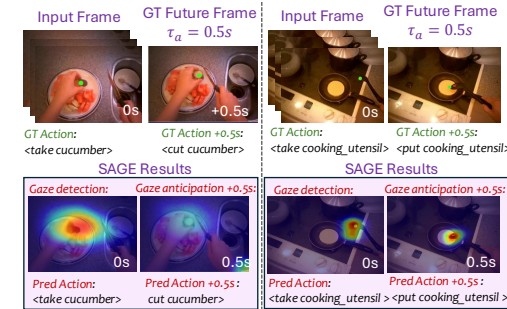

Figure 3: SAGE results on **Exo-Cook**, including action and gaze predictions for current and future frames.

Figure 4: SAGE results on **EGTEA Gaze+**, including action and gaze predictions for current and future frames.

Table 2: SAGE results on Exo-Cook dataset. [†]adapted for gaze anticipation

| Method | $\tau_a$ | Gaze Detection | | | HOI Detection | | | | | Gaze Anticipation | | | HOI Anticipation | | | | |
|---|---|---|---|---|---|---|---|---|---|---|---|---|---|---|---|---|---|
| | | F1 | Rec | Prec | mAP | Rec | Prec | Acc | F1 | F1 | Rec | Prec | mAP | Rec | Prec | Acc | F1 |
| VideoAttn Chong et al. (2020) | - | 55.6 | 72.8 | 54.8 | - | - | - | - | - | - | - | - | - | - | - | - | - |
| Sharingan Tafasca et al. (2024) | - | 56.6 | 72.8 | 55.3 | - | - | - | - | - | - | - | - | - | - | - | - | - |
| ST-Gaze Ni et al. (2023) | - | - | - | - | 42.5 | 77.0 | 61.9 | 59.0 | 66.9 | - | - | - | - | - | - | - | - |
| VideoAttn Chong et al. (2020)[†] | - | - | - | - | - | - | - | - | - | 42.2 | 54.6 | 42.5 | - | - | - | - | - |
| Sharingan Tafasca et al. (2024)[†] | - | - | - | - | - | - | - | - | - | 40.1 | 52.7 | 40.4 | - | - | - | - | - |
| ST-Gaze Ni et al. (2023) | 1 | - | - | - | - | - | - | - | - | - | - | - | 41.1 | 76.2 | 58.2 | 53.7 | 66.1 |
| SAGE (VideoAttn) | 1 | 55.8 | 73.2 | **55.7** | 44.2 | 79.0 | 62.6 | 58.5 | 68.9 | 48.5 | 62.1 | 49.6 | 42.4 | 76.6 | 59.0 | 54.6 | 66.9 |
| SAGE (Sharingan) | 1 | **57.7** | **73.4** | 55.5 | **44.6** | **79.4** | **63.1** | **60.2** | **69.4** | **49.2** | **62.8** | **50.0** | **42.5** | **76.8** | **59.2** | **54.8** | **67.2** |

the loss is similarly defined and summed over intermediate frames, $\mathcal{L}_{\text{act,2}} = -\sum_{k=t+1}^{t+M} y_k^{gt} \log \hat{y}_k$. The **Total Loss** function for training SAGE model formulates as $\mathcal{L} = \lambda_1 \mathcal{L}_{hm,1} + \lambda_2 \mathcal{L}_{hm,2} + \lambda_3 \mathcal{L}_{io} + \lambda 4 \mathcal{L}_{act,1} + \lambda_5 \mathcal{L}_{act,2}$.

## 5 EXPERIMENTS

**Datasets & Metrics.** To evaluate our framework, we use three datasets. **Vid-HOI** Chiou et al. (2021), an exocentric dataset for video-based HOI detection and anticipation, provides annotated sequences of human–object interactions. **EGTEA Gaze+** Li et al. (2018a), widely used for egocentric gaze and action tasks, contains over 28 hours of video across 86 action classes from 32 participants. **Exo-Cook**, introduced in this work, is the first exocentric dataset curated for joint HOI and gaze analysis. Following Ni et al. (2023), we evaluate HOI detection and anticipation on Vid-HOI and Exo-Cook using mean average precision (mAP), top-5 recall, precision, accuracy, and F1-score. For egocentric action recognition and anticipation, we report mean class accuracy Roy et al. (2024), while gaze detection and anticipation are evaluated with F1-score, recall, and precision.

Table 3: SAGE results on Vid-HOI dataset for HOI Detection and Anticipation in Oracle mode. * includes word embedding module.

| Method | HOI Detection | | | | | | HOI Anticipation | | | | | |
|---|---|---|---|---|---|---|---|---|---|---|---|---|
| | $\tau_a$ | mAP | Rec | Prec | Acc | F1 | $\tau_a$ | mAP | Rec | Prec | Acc | F1 |
| STTran Cong et al. (2021) | - | 28.32 | - | - | - | - | 1 | 29.09 | **74.76** | 41.36 | 36.61 | 50.48 |
| | | | | | | | 3 | 27.59 | **74.79** | 40.86 | 36.42 | 50.16 |
| | | | | | | | 5 | 27.32 | **75.65** | 41.18 | 36.92 | 50.66 |
| ST-Gaze* Ni et al. (2023) | - | 38.46 | **73.62** | 59.16 | **53.76** | 60.57 | 1 | 35.71 | 71.28 | 59.38 | 51.06 | 62.06 |
| | | | | | | | 3 | 32.19 | 71.09 | 60.17 | 51.63 | 62.39 |
| | | | | | | | 5 | 32.30 | 70.67 | 58.99 | 50.79 | 61.59 |
| SAGE (Ours) | 1 | **38.65** | 72.44 | 59.22 | 52.22 | 61.96 | 1 | **37.71** | 72.88 | **60.24** | 51.86 | 62.72 |
| | 3 | 38.18 | 72.21 | **59.95** | 52.18 | **62.02** | 3 | **34.22** | 72.36 | **60.98** | 52.88 | 63.05 |
| | 5 | 38.13 | 72.01 | 59.92 | 52.18 | 61.88 | 5 | **32.64** | 71.96 | **59.88** | 51.46 | 62.14 |

Table 4: SAGE results on EGTEA Gaze+ dataset. [†]adapted for gaze anticipation

| Method | Gaze Detection | | | Action Recognition | | | | Gaze Anticipation | | | Action Anticipation |
|---|---|---|---|---|---|---|---|---|---|---|---|
| | F1 | Rec | Prec | S1 | S2 | S3 | Avg | F1 | Rec | Prec | Mean-Cls Acc |
| Gaze MLE Li et al. (2021) | 26.6 | 35.7 | 21.3 | - | - | - | - | - | - | - | - |
| Joint Learning Li et al. (2018b) | 34.0 | 42.7 | 28.3 | - | - | - | - | - | - | - | - |
| Attention Transition Huang et al. (2018) | 37.2 | 51.9 | 29.0 | - | - | - | - | - | - | - | - |
| I3D-R50 Feichtenhofer et al. (2019) | 40.9 | 57.2 | 31.8 | - | - | - | - | - | - | - | - |
| MViT Fan et al. (2021) | 43.0 | 57.8 | 35.4 | - | - | - | - | - | - | - | - |
| GLC Lai et al. (2024a) | 44.8 | 61.2 | 35.3 | - | - | - | - | - | - | - | - |
| I3D-2Stream Li et al. (2021) | - | - | - | 55.8 | 53.1 | 53.6 | 54.2 | - | - | - | - |
| R34-2Stream Sudhakaran & Lanz (2018) | - | - | - | 62.2 | 61.5 | 58.6 | 60.8 | - | - | - | - |
| SAP Wang et al. (2020) | - | - | - | 64.1 | 62.1 | 62.0 | 62.7 | - | - | - | - |
| GC-TSM Hao et al. (2022) | - | - | - | **66.5** | **66.1** | 62.6 | **65.1** | - | - | - | - |
| I3D-R50 Feichtenhofer et al. (2019)[†] | - | - | - | - | - | - | - | 34.3 | 46.5 | 29.6 | - |
| MViT Fan et al. (2021)[†] | - | - | - | - | - | - | - | 31.5 | 44.8 | 28.5 | - |
| CSTS-Visual Lai et al. (2024c) | - | - | - | - | - | - | - | 31.3 | 46.2 | 28.1 | - |
| GLC Lai et al. (2024a)[†] | - | - | - | - | - | - | - | 32.8 | 48.5 | 28.7 | - |
| AFFT Zhong et al. (2023) | - | - | - | - | - | - | - | - | - | - | 35.2 |
| AVT Girdhar & Grauman (2021a) | - | - | - | - | - | - | - | - | - | - | 35.2 |
| MF Patrick et al. (2021) | - | - | - | - | - | - | - | - | - | - | 56.9 |
| ORVIT-MF Herzig et al. (2022) | - | - | - | - | - | - | - | - | - | - | 57.2 |
| InAViT Roy et al. (2024) | - | - | - | - | - | - | - | - | - | - | 58.2 |
| LAVILA Zhao et al. (2023) | - | - | - | - | - | - | 81.8 | - | - | - | - |
| SAGE (Ours) | **46.8** | **62.1** | **36.8** | 66.4 | 65.4 | **63.1** | 65.0 | **37.0** | **54.9** | **32.7** | **58.4** |

Table 5: SAGE Ablations of GCSA and GCTP on EGTEA Gaze+ Dataset

| Models | Gaze Detection | | | Action Recogn. | Gaze Anticipation | | | Action Anticipation |
|---|---|---|---|---|---|---|---|---|
| | F1 | Rec | Prec | Acc | F1 | Rec | Prec | Acc |
| SAGE-1 | 44.8 | 61.2 | 35.3 | - | - | - | - | - |
| SAGE-2 | - | - | - | 63.1 | - | - | - | - |
| SAGE-12 | 46.3 | 61.9 | 36.4 | 63.9 | | | | |
| SAGE | **46.8** | **62.1** | **36.8** | **65.0** | **37.0** | **54.9** | **32.7** | **58.4** |

In the below experiments, SAGE denotes the full model with GCSA and GCTP; SAGE-1 the gaze detection module; SAGE-2 the action recognition module; and SAGE-12 their combination with GCSA.

## 5.1 SAGE ON EXOCENTRIC VIDEOS

We evaluate SAGE on two exocentric datasets—Vid-HOI and Exo-Cook—and construct multiple adapted baselines from existing methods to enable a comprehensive comparison across all four tasks. **Exo-Cook Dataset.** We evaluate SAGE on four tasks, as summarized in Table 2. Since no existing methods directly benchmark on this dataset, we create multiple baseline models by adapting existing models on Exo-Cook. We incorporate two gaze detection model VideoAttn Chong et al. (2020) and Sharingan Tafasca et al. (2024) into SAGE architecture, noted as SAGE (VideoAttn) and SAGE (Sharingan). Correspondingly, VideoAttn and Sharingan are trained & evaluated on Exo-Cook as baselines for gaze detection and anticipation. We run ST-Gaze model Ni et al. (2023) on Exo-Cook to establish the first baseline for HOI detection and anticipation. SAGE (Sharingan) outperforms ST-Gaze on HOI detection by +2.1 mAP and +1.0 accuracy and achieves highest F1 score (69.4%). Similarly, on HOI anticipation, SAGE (Sharingan) has the best overall performance, with an F1 score of 67.2% and mAP of 42.5—surpassing the ST-Gaze baseline (F1: 66.1%, mAP: 41.1). In the gaze detection task, SAGE (Sharingan) achieves the highest F1 score (57.7%), slightly outperforming the best baseline Sharingan (56.6%) by +1.1 points. While Precision is similar (55.5 vs. 55.3), SAGE shows a modest gain in Recall (+0.6). For gaze anticipation, SAGE (Sharingan) achieves highest F1 score of 49.2%, Recall of 62.8, and Precision of 50.0, outperforming the closest baseline (ST-Gaze). By comparing SAGE (VideoAttn) and SAGE (Sharingan), we show that integrating more advanced gaze model architecture helps improve the joint model performances. We show some qualitative results in Figure 3, and provide more detailed qualitative results in the Appendix (Section E).

**Vid-HOI Dataset.** We take ST-Gaze Ni et al. (2023) and STTran Cong et al. (2021) as baseline models for HOI detection and anticipation in Table 3. Gaze is *not* evaluated on Vid-HOI as no gaze labels are provided. For HOI detection, our full model (SAGE) achieves the best mAP, Precision and F1 values. The improved performance of ST-Gaze* Ni et al. (2023) in Recall and Accuracy is attributed to the use of word embeddings as an additional input modality. We futher analyze the impact of Gaze-Conditioned-Attention in the spatial domain in the Appendix (Section D.2). For HOI anticipation, SAGE outperforms STTran Cong et al. (2021) and ST-Gaze Ni et al. (2023) across all anticipation horizons (1s, 3s, 5s) for mAP, Precision, Accuracy and F1.

## 5.2 SAGE ON EGOCENTRIC VIDEOS

We conduct a comprehensive evaluation on the EGTEA Gaze+ dataset Li et al. (2018a) in Table. 4, covering four tasks as described below.

**Gaze Detection & Anticipation.** We compare our gaze estimation performance against existing egocentric gaze estimation models, including GLC Lai et al. (2024a), MViT Fan et al. (2021), and I3D-R50 Feichtenhofer et al. (2019). As shown in Table 4, SAGE achieves the best F1 score of 46.8, significantly surpassing GLC (44.8), MViT (43.0), and I3D-R50 (40.9). It also achieves the highest Recall at 62.1 and the top Precision at 36.8. We improve GLC with 0.9% and 1.5% gain in Recall and Precision, respectively. For gaze anticipation task, as no existing baseline exists on EGTEA Gaze+, we established four baseline models by re-training GLC, MViT, I3D-R50 and CSTS-Visual Lai et al. (2024c)) with ground-truth future gaze heatmaps. In the sixth row of Table. 4, our method achieves the higher F1 score (37.0), Recall (54.9), and Precision (32.7) than four of the baseline models. These results highlight the advantage of integrating gaze modeling with action understanding under joint training framework. SAGE can serve as a strong baseline model for gaze anticipation on EGTEA Gaze+. We also study the sensitivity of SAGE to gaze modules in Appendix (Section D.1).

**Action Recognition.** In Table 4, we compare SAGE performances with other state-of-the-art methods on the EGTEA Gaze+ dataset in terms of average accuracy across three test splits. On the third test set (S3), SAGE achieves an accuracy of 63.1%, surpassing the performance of the GC-TSM model Hao et al. (2022), which reports 62.6%. Overall, SAGE obtains an average accuracy of 65.0%, which is comparable to GC-TSM's top score of 65.1%, while outperforming other recent baselines such as SAP(62.7%) and R34-2Stream (60.8%). Since SAGE does not leverage LLMs or large-scale external pretraining, we exclude LAVILA Zhao et al. (2023)—which uses LLMs and is pretrained on Ego4D+WIT—from direct comparison, but still report it in Table 4 for completeness.

**Action Anticipation.** We perform action anticipation with the full SAGE model for a future time gap of 0.5 seconds. SAGE outperforms prior models including AVT Girdhar & Grauman (2021a), and InAViT Roy et al. (2024), achieving the highest mean-class accuracy of 58.4%, surpassing the latest work InAViT (58.2%). Compared to transformer-based methods such as MF (56.9%) and AVT (35.2%), SAGE demonstrates a clear advantage in capturing gaze-conditioned temporal dependencies for anticipation. We show some qualitative results on EGTEA Gaze+ in Figure. 4.

**Ablation Study.** Table 5 presents the ablation study of the GCSA and GCTP modules on the EGTEA Gaze+ dataset. GCSA combines two independent models, SAGE-1 and SAGE-2, into a joint model SAGE-12, yielding improvements in both gaze detection and action recognition. This confirms that jointly learning gaze and action features with spatial attention enhances representations for both tasks. Further adding the GCTP module introduces temporal modeling for anticipation, resulting in further gains across all tasks and demonstrating GCTP's effectiveness in capturing predictive temporal dynamics. SAGE not only delivers strong accuracy but also remarkable efficiency, processing a 20-frame input sequence in just 0.63s on an NVIDIA RTX A6000, simultaneously performing gaze detection, action recognition, gaze anticipation (+0.5s), and action anticipation (+0.5s).

## 6 CONCLUSION

SAGE integrates simultaneous recognition and anticipation of human actions and gaze within a unified end-to-end trainable model, using our GCSA and GCTP modules. Its modular design enables viewpoint-specific training across egocentric and exocentric datasets, which is validated through extensive evaluation on three benchmark datasets: VidHOI, EGTEA Gaze+ and our newly proposed Exo-Cook dataset. Our results demonstrate that synergy between gaze and actions in the current and future frames in SAGE compares favorably and even outperforms individual task specialized state-of-the-art models.

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

# A    IMPLEMENTATION DETAILS

## A.1    HUMAN-OBJECT-PAIR FEATURE EXTRACTION

We refer to ST-Gaze Ni et al. (2023) to pre-process Vid-HOI and Exo-Cooking datasets. An object module is applied to process the video sequence and detects $N$ bounding boxes $\{b_{t,j}\}$ along with their corresponding classes. Human bounding boxes are also detected. In addition, human head are detected by YOLO-v5 model Jocher et al. (2022) and paired with their body bounding boxes. DeeoSort model is applied to associate these detections with previous detections to establish trajectories for each detected human $\{H_i\}$ and object $\{O_j\}$. Video feature are extracted by ResNet.

## A.2    MODEL ARCHITECTURE

The spatial encoder in HOI recognition model consists of 3 Transformer encoder layers. The gaze anticipation model consists of 3 temporal Transformer layers, each layer contains one Multi-head self-attention block and one cross-attention block. We apply the Transformer decoder from GLC Lai et al. (2024a) for producing gaze heatmaps. The HOI anticipation module (GCTP) consists of 3 temporal layers

## A.3    TRAINING SETTINGS

Based on the complexity of the full SAGE model and the difficulty in training for all the tasks. We initialize some models with pre-trained model and multi-stage training process.

- The Gaze detection module is initialized with the pre-trained model from VideoAttn Chong et al. (2020) (on Vid-HOI dataset) or GLC Lai et al. (2024a).
- We first train joint model SAGE-12 for 5 epochs with gaze labels and action labels for the current sequence.
- Then we load the weights from SAGE-12 and train the full model SAGE with future gaze and action annotations. We train 25 epochs in total.
- In the loss function $L = \lambda_1 L_{hm,0:t} + \lambda_2 L_{hm,t+1:t+M} + \lambda_3 L_{io} + \lambda 4 L_{act,t} + \lambda_5 L_{act,t+1:t+M}$, we set $\lambda_1 = 1.0, \lambda_2 = 1.0, \lambda_3 = 0.5, \lambda_4 = 1.5, \lambda_5 = 1.5$.

# B    EXO-COOK BENCHMARK CREATION

We construct a small benchmark based on a subset from Ego-Exo 4D. We describe more details for pre-processing and generating the labels in this section. On cooking videos, we try BERT model Devlin et al. (2019) to generate sentence embedding and cluster text into action categories. We extract the text descriptions from Ego-Exo 4D cooking videos and generate 30250 video clips and the corresponding action descriptions. We generate text embeddings for each video clip and apply K-means algorithm to perform action clustering based on the semantic meaning of the text.

## B.1    A. GAZE LABEL CREATION

Ego-Exo 4D provide raw gaze annotations from the eye-tracking device (Project Aria). To generate 2D gaze labels for third-person views in Exo-Cook, we follow a calibrated projection pipeline that transforms the 3D gaze direction from the eye-tracking device (Project Aria) into 2D image coordinates of external cameras.

**A.1 Gaze Ray Extraction.**    For each frame, we extract the 3D gaze origin and direction vectors from the left and right eyes using Aria's 'scene gaze' and 'eye gaze' streams. These are defined in the Aria scene camera coordinate system and describe binocular gaze rays:

$$\text{Ray}_L = (\boldsymbol{o}_L, \boldsymbol{d}_L), \quad \text{Ray}_R = (\boldsymbol{o}_R, \boldsymbol{d}_R)$$

**A.2 3D Gaze Estimation in Aria Frame.**    Each frame of Project Aria includes left and right eye gaze vectors with respect to the device's coordinate frame. We estimate the 3D point of gaze $\boldsymbol{G}_{3D}^{\text{aria}}$ by computing the 3D intersection (or midpoint approximation) of the two gaze rays $\text{Ray}_L$ and $\text{Ray}_R$.

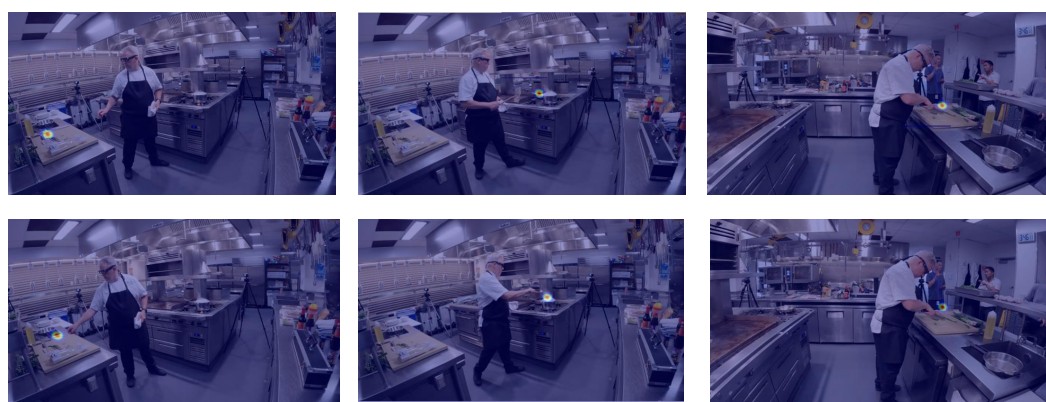

Figure 5: Individual examples (six) of created ground-truth gaze heatmap from Exo-Cook.

**A.3 Coordinate Transformation.**    To localize the gaze in the third-person camera frame, we use the known extrinsic calibration between the Aria device and each external camera. Let $\boldsymbol{T}_{\text{aria}\rightarrow\text{cam}}$ denote the 6DoF pose of the Aria glasses with respect to a third-person camera. We transform the 3D gaze point into the third-person coordinate system via:

$$\boldsymbol{G}_{3D}^{\text{cam}} = \boldsymbol{T}_{\text{aria}\rightarrow\text{cam}} \cdot \boldsymbol{G}_{3D}^{\text{aria}}$$

**A.4 3D-to-2D Projection.**    Given the intrinsic camera matrix $\boldsymbol{K}$ of the third-person camera and the transformed gaze point $\boldsymbol{G}_{3D}^{\text{cam}}$, we project the gaze to the image plane:

$$\boldsymbol{G}_{2D}^{\text{cam}} = \Pi(\boldsymbol{K}, \boldsymbol{G}_{3D}^{\text{cam}})$$

where $\Pi$ denotes the standard perspective projection. This gives us the 2D gaze fixation point in pixel coordinates.

**A.5 Heatmap Generation.**    Following prior work Chong et al. (2020), we convert each gaze fixation point into a 2D Gaussian heatmap $\boldsymbol{M}^{\text{pseudo}} \in \mathbb{R}^{H \times W}$. The heatmap is centered at $\boldsymbol{G}_{2D}^{\text{cam}}$ with an isotropic Gaussian kernel:

$$\sigma = \frac{W_{\text{hm}} + H_{\text{hm}}}{2} \cdot \frac{3}{64}$$

where $W_{\text{hm}}$ and $H_{\text{hm}}$ are the dimensions of the heatmap. For a $64 \times 64$ heatmap, this yields $\sigma = 3$ pixels. We show examples of generated gaze heatmaps in Figure. 5.

To handle occlusions or failed triangulation cases, we discard gaze points where the intersection error between eye rays exceeds a threshold. Additionally, all transformations and projections are timestamp-aligned to ensure synchronization between Aria and third-person views.

## B.2   B. ACTION LABEL CREATION

**B.1 Action Description Alignment**    Each atomic description is associated with the following key metadata:

- `start_frame` and `end_frame`: specifying the frame-level temporal boundaries of the described action.
- `source_view`: indicating which third-person or egocentric camera the annotation corresponds to.
- `text`: raw textual description of the action.

To align each action with global time stamp, we use the provided frame indices and convert them to time given the video frame rate. We first sample the video frames near the `start_frame` for action recognition. Then we define a time offset $\tau_a = 1s$ beyond the `end_frame` to define the target for action anticipation.

Table 6: Verb-object extraction and semantic object selection using spaCy and BERT. In the third column, we show the cosine similarity between the spaCy object and the object name mentioned in Exo-Cook raw annotation.

| Sentence | Verb, Object | spaCy Object List | Object Name |
|---|---|---|---|
| C places the spoon in the bowl in his right hand. | places, spoon | spoon (0.652), bowl (1.000), hand (0.437) | Bowl |
| C slices the garlic on the chopping board with his right hand. | slices, garlic | garlic (0.424), board (0.869), hand (0.341) | Chopping board |
| C adds the sliced garlic into the frying pan with his right hand. | adds, garlic | garlic (0.461), pan (1.000), hand (0.378) | Pan |
| C adds oil into the frying pan with the oil bottle in his left hand. | adds, oil | oil (0.552), pan (0.449), bottle (1.000), hand (0.437) | Bottle |

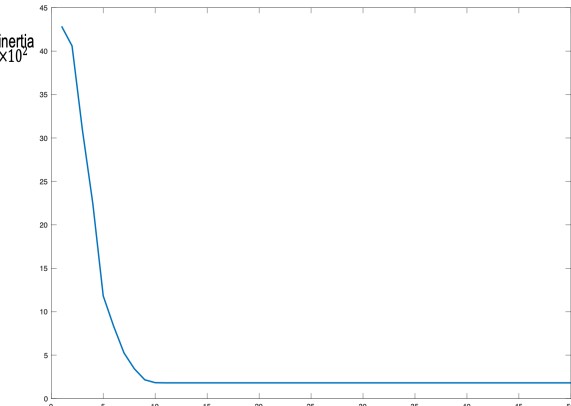

Figure 6: The inertia plot of the clustering for all the text embeddings from cooking video clips. We set K = 10 to generate 10 categories for the actions.

**B.2 Action Simplification with spaCy**    We use spaCy Explosion (2024) to perform syntactic parsing and extract verb-object pairs from each sentence. For example, given the description *"C slices the garlic on the chopping board with his right hand"*, spaCy identifies `slices` as the verb and `garlic` as the primary object. In more complex sentences containing multiple noun phrases (e.g., "C adds the sliced garlic into the frying pan with his right hand"), spaCy is used to list all candidate objects (e.g., `garlic`, `pan`, `hand`). To identify the most semantically relevant object, we compute the cosine similarity between the BERT Devlin et al. (2019) embedding of each noun and the embedding of the full sentence. The object with the highest similarity is selected as the primary target. For instance, in the garlic example, `pan` receives the highest score, thus being identified as the object most aligned with the described action context. In Table. 6, we show the Cosine Similarity between different sentences extracted from the annotation file of Ego-Exo 4D.

**B3 Action Clustering**    As described in Section B.1 and B.2, we repeat the action description alignment and simplification process for all the samples we cropped from Exo-Cook. After applying spaCy for extracting essential semantic components for each clip, we employ the pre-trained BERT Devlin et al. (2019) model to transform these extracted components into high-dimensional text embeddings. Once text embeddings for all sequences are obtained, we apply $K$-means clustering to group similar embeddings. To determine the optimal number of clusters, we use the Elbow Method by analyzing the inertia, defined as the sum of squared distances between each data point and the centroid of its assigned cluster. The inertia values across different choices of $K$ are illustrated in Figure 6. Based on this analysis, we set the number of clusters to $K = 10$ and therefore, we have 10 categories of actions in Exo-Cook.

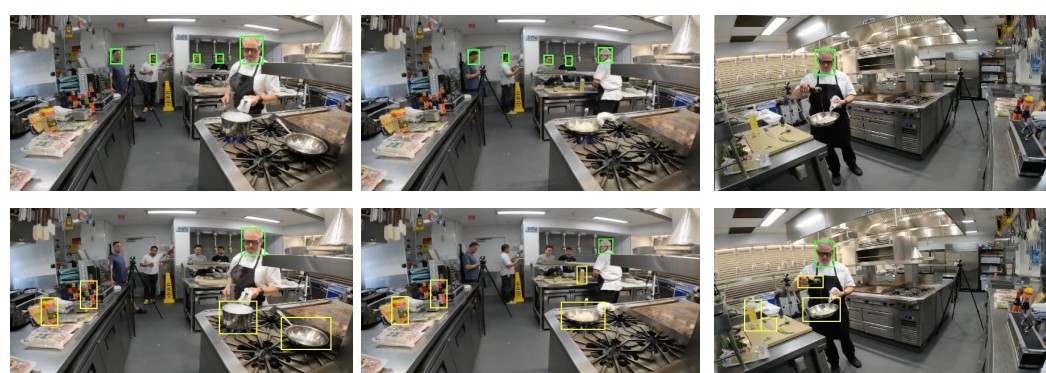

Figure 7: Examples (three) of bounding box detection for human head and object in Exo-Cook images. The first row represents head bounding box (green color) detection for all subjects in the scene. The second row shows the head bounding box for the key participant and the objct bounding boxes (yellow color).

### B.3   C. Bounding Box for Human and Object

We follow the pipeline in Ni et al. (2023) and use YOLOv5 Jocher et al. (2022) to detect full-body and head bounding boxes of the camera wearer in third-person views.

**C.1 Bounding Box Detection**   The first row of Figure 7 illustrates examples of head bounding box detections (highlighted in green) within Exo-Cook images, capturing every person appeared in the scene. Similarly, the second row presents object bounding box detections, marked in yellow, covering all relevant objects within the scene.

**C.2 Bounding Box Matching**   As our primary interest is the activity of the "major participant" equipped with the Project Aria device, we implement bounding box matching based on the projected location of the device. Specifically, the head bounding box containing the 2D projection of the Project Aria's center is designated as the major participant. Consequently, as depicted in the second row of Figure 7, any unrelated head bounding boxes will be removed.

## C   Baseline Model for Gaze Anticipation

### C.1   Exocentric View

Due to lack of baseline methods on Gaze Anticipation task, we propose to adapt existing models as baseline. In Table 1 of the main paper, we evaluate gaze anticipation performance on the Exo-Cook dataset by adapting two baseline models: VideoAttn Chong et al. (2020) and Sharingan Tafasca et al. (2024). These baseline models, initially designed for gaze detection tasks, were modified and retrained to predict gaze heatmaps at a future timestamp (1 second ahead) based solely on the last frame of a given sequence. The results show the advantage of employing the temporal modeling in our SAGE model. Specifically, our temporal-aware SAGE significantly outperforms the adapted baseline models across all gaze anticipation metrics. For instance, the SAGE model (with Sharingan as its gaze detection backbone) achieves an F1 score of 49.2, notably higher than adapted VideoAttn (42.2) and adapted Sharingan (40.1). Similarly, recall (62.8 vs. 54.6 and 52.7) and precision (50.0 vs. 42.5 and 40.4) metrics also show substantial improvements.

### C.2   Egocentric View

Similarly, we show comparison results in Table 4 of the main paper. CSTS Lai et al. (2024c) is the state-of-the art model for egocentric gaze anticipation. However, CSTS Lai et al. (2024c) is not trained/evaluated on EGTEA Gaze+ as they require audio data. We adapt their backbone model, denoted as "CSTS-Visual Lai et al. (2024c)" in Table. 4 to gaze anticipation task on EGTEA Gaze+.

Table 7: Sensitivity of SAGE to gaze modules on the EGTEA Gaze+ dataset

| Model | Gaze Model | Gaze Detection | | | Action Recogn. | Gaze Anticipation | | | Action Anticipation |
|---|---|---|---|---|---|---|---|---|---|
| | | F1 | Rec | Prec | Top-1 Acc | F1 | Rec | Prec | Mean-cls Acc |
| SAGE | I3D-R50 Feichtenhofer et al. (2019) | 42.2 | 58.5 | 33.2 | 64.6 | 34.7 | 47.4 | 30.2 | 57.8 |
| | MViT Fan et al. (2021) | 44.5 | 58.8 | 35.5 | 64.6 | **38.8** | 54.2 | 32.5 | 58.0 |
| | GLC Lai et al. (2024a) | **46.8** | **62.1** | **36.8** | **65.0** | 37.0 | **54.9** | **32.7** | **58.4** |

Table 8: Top-1 accuracy for SAGE at longer anticipation horizon on the EGTEA Gaze+ dataset

| Method | $\tau_a$ | | | |
|---|---|---|---|---|
| | 0.5 | 1.0 | 1.5 | 2.0 |
| InAViT Roy et al. (2024) | 67.8 | 66.9 | 65.8 | 64.1 |
| SAGE (Ours) | **68.0** | **67.3** | **66.2** | **64.5** |

In addition to CSTS-Visual, we adapt another three existing models for gaze anticipation, denoted as GLC[†] Lai et al. (2024a), MViT[†] Fan et al. (2021) and I3D-R50[†] Feichtenhofer et al. (2019). These models were retrained to predict gaze heatmaps at 0.5 seconds into the future. We re-train the four models by using the grounf-truth heatmaps at the 0.5 seconds later. However, their performance was substantially lower compared to SAGE.

# D  EXTENSIVE EXPERIMENTAL ANALYSIS

## D.1  RESULTS ON EGTEA GAZE+

In the main paper, we compare the performance of SAGE with state-of-the-art models across different task domains. In the exocentric view, the effectiveness of gaze is more apparent, as the gaze model can explicitly identify the visual target of the person within the scene. Intuitively, the gaze-based attention closely aligns with the object being interacted with, making it a strong prior for modeling human-object interactions. In Table 1 of the main paper, we show the sensitivity of SAGE performance to the accuracy of gaze model. We incorporate two different gaze backbone model, denoted as SAGE (Gaze Detection Model: VideoAttn) and SAGE (Gaze Detection Model: Sharingan) in Table 1, into SAGE architecture. The results show that better gaze detection model in SAGE enhances performances for other three tasks.

Similarly, to explore the sensitivity of SAGE to the quality of gaze estimation in egocentric videos, we compare three different gaze backbones in Table 7: I3D-R50 Feichtenhofer et al. (2019), MViT Fan et al. (2021), and GLC Lai et al. (2024a). We observe that improvements in gaze detection performance consistently lead to better action recognition and anticipation results. For instance, GLC achieves the best gaze detection F1 score (46.8), which corresponds to the highest top-1 accuracy in action recognition (65.0) and action anticipation (58.4). In contrast, models with weaker gaze detection performance, such as I3D-R50 (F1 = 42.2), yield the lowest accuracy in both action recognition and anticipation. These results suggest that precise gaze localization is crucial for enhancing action detection and anticipation in egocentric settings. Intuitively, gaze reflects the subject's focus of interest and serves as an informative prior for modeling human-object interactions. The results in Table 7 prove that better gaze attention enhances the performance for each task. It validates the effectiveness of our GCSA module.

Following InAViT Roy et al. (2024), we explore SAGE performance on longer time horizons. In Table 8, we train SAGE on different time horizons {0.5s, 1s, 1.5s, 2s}. Notably, SAGE consistently outperforms InAViT on all time horizons. As the anticipation horizon increases, there are greater uncertainty for future action prediction. The Top-1 accuracy of action anticipation in InAViT decreases from 67.8 to 64.1 when the anticipation time increases to 2.0s. As SAGE is anticipating future actions with multiple intermediate time stamps (i.e., $[y_{t+1}, \cdots, y_{t+M}]$), SAGE maintains more stable performance over longer time spans.

Table 9: HOI detection in Oracle mode on Vid-HOI dataset Chiou et al. (2021). * includes word embedding module.

| Method | $\tau_a$ | mAP | Rec | Prec | Acc | F1 |
|---|---|---|---|---|---|---|
| STTran Cong et al. (2021) | - | 28.32 | - | - | - | - |
| ST-Gaze* Ni et al. (2023) | - | 38.46 | **73.62** | 59.16 | **53.76** | 60.57 |
| ST-Gaze Spatial* Ni et al. (2023) | - | 36.29 | 71.03 | 59.38 | 51.72 | 61.24 |
| SAGE-12 (Ours) | - | **38.08** | 72.11 | 59.90 | 52.12 | **62.03** |
| SAGE (Ours) | 1 | **38.65** | 72.44 | 59.22 | 52.22 | 61.96 |
| | 3 | 38.18 | 72.21 | **59.95** | 52.18 | 62.02 |
| | 5 | 38.13 | 72.01 | 59.92 | 52.18 | 61.88 |

Table 10: SAGE Ablations of GCSA (SAGE-12) and GCTP (SAGE) on Exo-Cook Dataset

| Models | Gaze Detection | | | HOI Detection | Gaze Anticipation | | | HOI Anticipation |
|---|---|---|---|---|---|---|---|---|
| | F1 | Rec | Prec | Acc | F1 | Rec | Prec | Acc |
| SAGE-1 | 56.6 | 72.8 | 55.3 | - | - | - | - | - |
| SAGE-2 | - | - | - | 58.9 | - | - | - | - |
| SAGE-12 | 57.5 | **73.6** | **56.0** | 59.9 | | | | |
| SAGE | **57.7** | 73.4 | 55.5 | **60.2** | 49.2 | 62.8 | 50.0 | **54.8** |

## D.2 RESULTS ON VID-HOI

In Table 9, we benchmark HOI detection performance on the Vid-HOI dataset using ST-Gaze Ni et al. (2023) and STTran Cong et al. (2021) as baselines. Since Vid-HOI does not provide gaze labels, gaze evaluation is omitted. To specifically evaluate the impact of Gaze-Conditioned Attention in the spatial domain, we introduce a spatial-only variant of ST-Gaze by removing its temporal layer (referred to as ST-Gaze Spatial). Additionally, we include SAGE-12 in the comparison. Notably, SAGE-12 achieves the highest F1 score (62.03), underscoring the importance of integrating gaze-based attention mechanisms into HOI recognition tasks. Furthermore, our full SAGE model attains the best mAP (38.65) and Recall (72.44) scores at the 1-second anticipation setting, and the highest Precision (59.95) at the 3-second setting. It should be noted that due to space constraints, the performances of ST-Gaze Spatial and SAGE-12 were inadvertently omitted from the main paper (Table 2); we present their comprehensive results here for clarity. We also corrected the boldface labeling in the "Rec" and "Acc" column. In the main paper (Table 2), the value 72.44 and 52.22 was incorrectly highlighted in bold. This has been corrected by appropriately emphasizing 73.62 as the Recall value and 53.76 as the Accuracy value for ST-Gaze*, which outperforms SAGE. This improved performance is attributed to ST-Gaze* use of word embeddings as an additional input modality,

## D.3 SAGE ABLATION STUDY ON EXO-COOK

In the main paper, we show ablation study of model combination on an egocentric dataset, i.e. EGTEA Gaze+ dataset. We conduct similar experiments on an exocentric dataset, i.e. Exo-Cook dataset. In Table 10, we show how each component in our model contribute to the final performances through ablation study. We study the performance of different combinations of the four modules and how they affect the performance of each other. We use Sharingan Tafasca et al. (2024) as the gaze model.

- **SAGE-1:** the model for gaze heatmap prediction (Sharingan Tafasca et al. (2024) as backbone.

- **SAGE-2:** single model for action detection

- **SAGE-12:** joint model for gaze and HOI detection, we integrate GCSA between SAGE-1 and SAGE-2.

- **SAGE:** final joint model that integrate all the components. GCSA and GCTP modules are integrated.

SAGE-1 and SAGE-2 perform gaze detection and HOI detection (oracle mode) independently and we observe that combining them into the joint model SAGE-12 leads to consistent improvements in both tasks. The performance improvements highlight the contribution of the GCSA module, which effectively integrates gaze into spatial attention, allowing the model to better capture human-object interactions. Furthermore, when incorporating the GCTP module into the full SAGE model, which explicitly models gaze-conditioned temporal relationships, we observe additional improvements for gaze detection and HOI detection . The results shows that incorporation via GCTP enhances the model's ability to learn temporal correlations and reason about gaze and human actions. The full SAGE model can predict future gaze and HOI based on the encoded temporal cues. It should be noted that as we select the optimal SAGE based on HOI detection result, the Gaze Detection performance might fluctuate around the optimal. In Table 5, the F1 and Rec of SAGE slightly go down compared to SAGE-12. This also indicates that gaze can be less reliable in exocentric videos than egocentric videos, which is understandable. Overall, the progressive improvements from SAGE-1/2 to SAGE-12 and finally to the full SAGE model demonstrate that both GCSA and GCTP are essential for fully leveraging the bidirectional relationship between gaze and HOIs in both current and future frames.

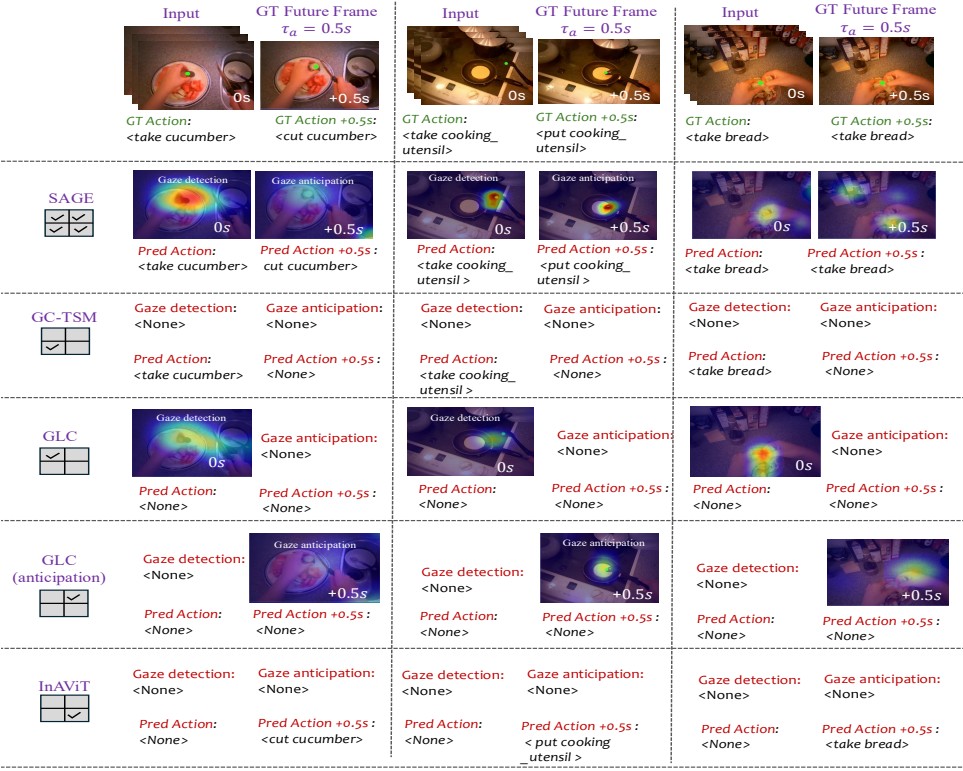

Figure 8: Visualization of SAGE results on the EGTEA Gaze+ dataset, illustrating action and gaze estimation for both current and future frames. The leftmost column displays checkbox representation corresponding to each method, indicating the output settings available. Our method (SAGE) supports all four prediction settings: gaze detection, action recognition, gaze anticipation, and action anticipation.

# E   QUALITATIVE RESULTS OF SAGE

Figure 8 provides qualitative comparisons of our proposed model **SAGE** with four competitive baselines—GC-TSM Hao et al. (2022), GLC Lai et al. (2024a), GLC (anticipation), and InAViT Roy

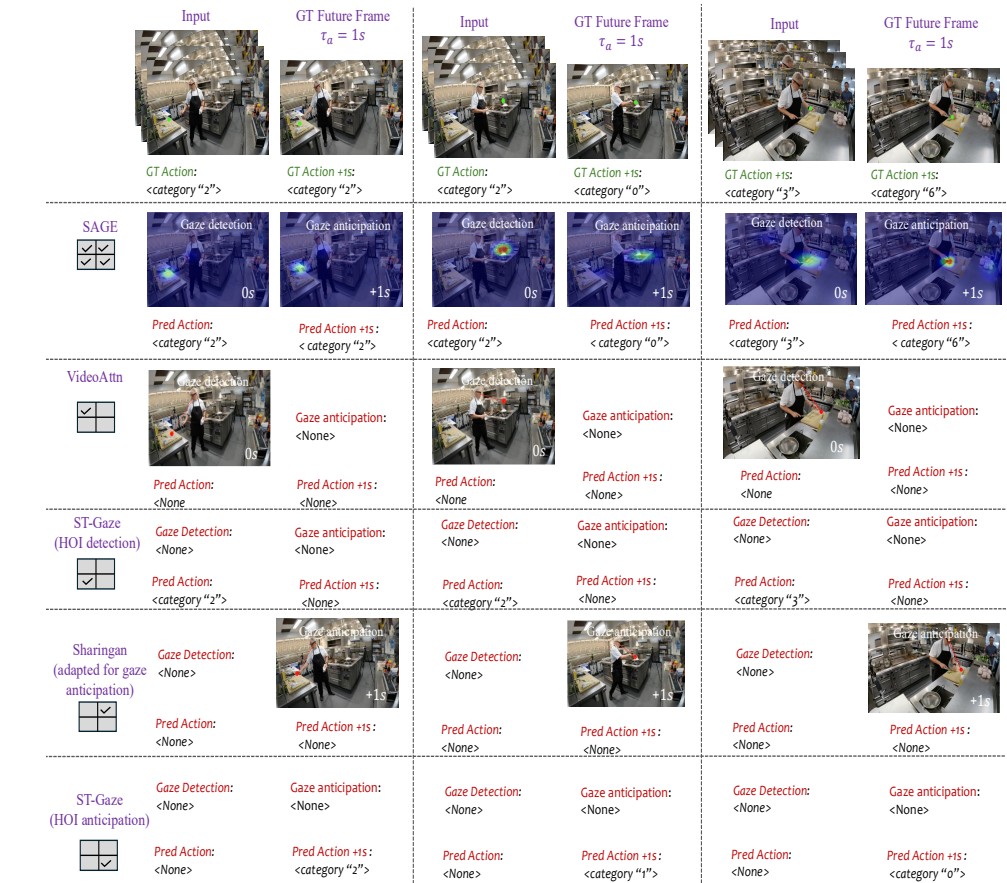

Figure 9: Visualization of SAGE results on Exo-Cook, including action and gaze results for current and future frames. The leftmost column displays checkbox representation corresponding to each method, indicating the output settings available. Our method (SAGE) supports all four prediction settings: gaze detection, action recognition, gaze anticipation, and action anticipation.

et al. (2024)—on the EGTEA Gaze+ dataset. The evaluation covers four tasks: gaze detection, gaze anticipation, action recognition, and action anticipation, with an anticipation horizon of $\tau_a = 0.5$ seconds. Each column shows the last frame (at $0s$) in the input sequence and its corresponding ground-truth future frame at $+0.5s$. The left example illustrates a two-step activity transition from *taking a cucumber* to *cutting the cucumber*. The middle example captures the action progression from *taking a cooking utensil* to *putting it down*, while the right example depicts a temporally stable action—*taking bread*. SAGE can performs all four tasks. The gaze heatmap predicted from SAGE for current and future frame is well-aligned with the interaction object in the scene. In the meantime, SAGE can predict current and future actions. GC-TSM does not model gaze and only predicts the current action, failing to anticipate future actions. GLC produces accurate gaze detection at $0s$, while GLC (anticipation) provides plausible future gaze at $+0.5s$; however, neither model completes all four tasks jointly. Notably, both variants fail in action anticipation. InAViT does not model gaze and can predict future actions.

In Figure 9, we compare SAGE with four state-of-the-art models in different task domain—VideoAttn Chong et al. (2020), ST-Gaze Ni et al. (2023), and Sharingan Tafasca et al. (2024)—using identical input video sequences. Notably, ST-Gaze provides two separate pipelines for HOI detection and HOI anticipation. While these baseline models are typically designed for individual tasks such as gaze detection or HOI classification, SAGE is the only model can jointly performing all four tasks. The figure showcases three representative examples from the Exo-Cook dataset, displaying the last frame at the current time ($0s$) and a future frame at the anticipation horizon $\tau_a = 1s$. The leftmost example illustrates a stable action sequence, while the middle and right

examples involve action transitions. Since actions are labeled with categorical indices, we compare category predictions across models. As discussed in the main paper, no existing model is specifically designed for exocentric gaze anticipation; hence, we adapt Sharingan Tafasca et al. (2024) for this task. Sharingan is the SOTA model for exocentric gaze target detection in images. However, the adapted Sharingan fails to accurately predict future gaze heatmaps across all three examples. Meanwhile, ST-Gaze can perform either HOI detection or anticipation, but not both simultaneously, and it fails to predict the correct HOI category in the rightmost example. These results highlight the advantage of SAGE in jointly modeling spatial and temporal dynamics for comprehensive human activity understanding.

