# OpenReview forum: "SAGE: A Synchronized Action and Gaze Estimation Framework for Comprehensive Human Behavior Analysis"
_ICLR.cc/2026/Conference — ICLR 2026 Conference Withdrawn Submission_

### Official Review · Reviewer_U9pX · 2025-10-25

**Soundness:** 2
**Presentation:** 2
**Contribution:** 2
**Rating:** 2
**Confidence:** 4

**Summary:**

This paper proposes a unified framework for detection and anticipation of human-object interaction (action) and human gaze. A transformer-based architecture is used to incorporate current gaze data to predict current and future actions and gaze. The framework also supports both egocentric and exocentric videos as input, enabling broader applicability across diverse scenarios. In addition, a new benchmark is contructed to facilitate research of actions and gaze in exocentric videos.

**Strengths:**

1) A unified framework is presented which integrates recognition and anticipation of both human-object interaction (action) and gaze.
2) A gaze conditioned spatial attention (GCSA) module is proposed to provide human-object interaction cues in the spatial domain and a gaze conditioned temporal prediction (GCTP) module is proposed to model temporal correlations between future gaze patterns
and future actions.
3) A benchmark (Ego-Cook) is established to facilitate research of action and gaze in exocentric videos.

**Weaknesses:**

1) **Limited novelty**. At first I am interested to see a unified framework which tries to jointly optimize recognition and anticipation of actions and gaze. However, it turns out to be a straightforward transformer-based approach which detects gaze, recognizes actions, anticipate future gaze and then future actions in a sequential way. There is lack of novelty/insights within the so-called joint optimization of these four tasks. Besides, it might cause accumulation of error in the propose approach.
2) **Insufficient ablation study**. The paper's ablation study (Table 5) is not sufficient to validate the effectiveness of the proposed modules. Specifically, the comparison of SAGE-12 and SAGE can not validate the proposed GCTP, because SAGE-12 is simply not enabled with the function of anticipation. A more suitable way would be to compare with some baseline which enables anticipation. In addition, the F1 score (based on precision & recall) of gaze detection for SAGE-12 and SAGE is not correct. That is, F1 score of SAGE is 46.2 other than the reported 46.8.
3) **Lack of empirical support for cross-view adaptability**. Although it is claimed in the paper (Section 4.5) that the proposed framework is the first to generalize across egocentric and exocentric views and enables broader applicability across diverse scenarios, there is no empirical study (i.e., cross-viewpoint adaptability) to support such claim. In practice, the proposed framework is trained and used separately for the two views, and it is not convincing how this framework is superior than other single-view methods.

**Questions:**

1) The paper extracts visual features based on human-object pairs following Ni et al. (2023). The feature extraction process would be more complex in egocentric videos with cluttered objects, and it is not clear how to decide the largest number of human-object pairs in each sequence.
2) In Section 4.6, in-out loss is defined as the binary cross-entropy between the predicted in-out label and the ground truth label. However, I can not find any information regarding in-out loss in other parts of the paper.
3) The experiment setting for gaze anticipation (e.g., future time period) is not explained in the paper.
4) As far as I know, there is several prior work proposed for joint gaze estimation and action recognition in egocentric videos, however, which are not compared in Table 4. For example, "Huang et al., Mutual Context Network for Jointly Estimating Egocentric Gaze and Action" (TIP 2020).

**Details Of Ethics Concerns:**

NA.

---

### Official Review · Reviewer_7XsA · 2025-10-29

**Soundness:** 2
**Presentation:** 2
**Contribution:** 2
**Rating:** 4
**Confidence:** 4

**Summary:**

This paper proposes SAGE, a unified framework for the simultaneous recognition and prediction of Human-Object Interaction (HOI) and gaze behavior. The work integrates gaze data into a spatio-temporal attention mechanism, achieving joint prediction of current and future actions and gaze through the proposed GCSA and GCTP modules. The authors also construct the Exo-Cook benchmark dataset, filling a data gap for HOI and gaze analysis from a third-person perspective. Overall, this is a multi-modal behavior analysis framework with practical value, but it has certain shortcomings in justifying its novelty, experimental design, and writing clarity.

**Strengths:**

1. To address the lack of third-person video datasets for joint HOI and gaze analysis, the authors construct the Exo-Cook benchmark derived from Ego-Exo4D.

2. SAGE unifies four key tasks: gaze detection, action recognition, gaze prediction, and action prediction into a single end-to-end trainable framework.

**Weaknesses:**

1. There are no ablation studies to validate the chosen ratios of the loss hyperparameters (λ1, λ2, λ3, λ4, λ5), casting doubt on the rationale behind their specific configuration.

2. The presentation of the paper needs improvement; the layout of Figures 1 and 2, in particular, requires refinement for better clarity.

3. The related work is insufficient. Some key studies in the computer vision community on joint modeling or establishing correlations between action and gaze are not discussed.

4. The baseline methods used for experimental comparison are limited to those published before 2024, which weakens the claim to state-of-the-art relevance.

5. It is recommended to investigate the impact of different gaze prediction time horizons on action prediction performance. This analysis could provide valuable insights into the interplay between these tasks.

**Questions:**

Please refer to the above weaknesses.

**Details Of Ethics Concerns:**

None.

---

### Official Review · Reviewer_ey1L · 2025-10-31

**Soundness:** 3
**Presentation:** 2
**Contribution:** 3
**Rating:** 6
**Confidence:** 3

**Summary:**

The authors propose a framework called SAGE that performs gaze and action recognition and anticipation simultaneously, and argue that this leads to better and more comprehensive understanding of human behavior. The framework is designed to support analysis on both egocentric and exocentric data. The authors also introduce a dataset specifically intended for benchmarking models in exocentric scenarios.

**Strengths:**

- The task is clearly defined and valuable. By integrating the analysis of both action and gaze, it contributes to a better understanding of human behavior.  The proposed dataset benefits the community by facilitating research on this task.
- The model design is clear and concise, and the data collection process is relatively rigorous.

**Weaknesses:**

- The action labels in Exo-Cook are generated by clustering vectorized textual descriptions, but no further explanation or human validation of the ten categories are provided. Since the generation of  labels relies heavily on the BERT model and lacks human verification or supervision, it is difficult to determine whether the labeling process introduces bias, which might make the labels easier for models to classify, but be potentially harmful for human understanding and real-world usage.
- There are no results reported for different prediction horizons on EGTEA Gaze+ and Exo-Cook.
- Exo-Cook is limited to the cooking scenario and lacks validation across more diverse scenarios.
- Bad formats and presentation
	- All citations are not in parenthesis, which violates the instructions provided in the template pdf file.
	- The aspect ratio of Figure 1 & Figure 2 is strange.
	- It would be better if the four contibutions in the end of introduction are placed in `itemize` format.

**Questions:**

- In Table 3, for HOI detection, why does SAGE’s performance on current HOI prediction vary with different future prediction horizons? Section 4.2 states that only information from time steps 1 to t is used, and no information from >t is introduced.
- Provide a more detailed explanation and visualization of the action labels in Exo-Cook. Include examples for each label, with both images and textual descriptions. Additionally, manually annotate a small subset of the data and show the consistency between the manual annotations and the existing labels.
- Provide results for different prediction horizons on EGTEA Gaze+ and Exo-Cook.
- Fix the citation format that does not meet the required standards.

---

### Official Review · Reviewer_Yv4r · 2025-11-02

**Soundness:** 2
**Presentation:** 1
**Contribution:** 3
**Rating:** 2
**Confidence:** 4

**Summary:**

The paper introduces SAGE, a transformer-based framework for synchronized action and gaze estimation.
SAGE integrates spatial attention and temporal prediction modules to jointly model human-object interactions and gaze, advancing both current and future activity prediction in egocentric and exocentric video data. The paper also extracts a gaze-oriented cooking subset from Ego-Exo4D to produce a dataset suitable for studying joint gaze and action. Evaluation includes EGTEA Gaze+ and Vid-HOI datasets and extensive experiments against the literature, demonstrating some value in join gaze-action modeling.

**Strengths:**

- S1 - SAGE cohesively models gaze and human-object interaction (HOI) in both recognition and anticipation tasks, which does seem to be a new approach.  The modeling approach is modularized allowing the framework to be applied to different types of action oriented videos.

- S2 - The introduction of Exo-Cook to support the combined analysis is well received; it is not clear if all of the additional parameters computed on it will be shared, but nonetheless it is a contribution.

- S3 - The evaluations against the literature demonstrate the potential of the methods.

**Weaknesses:**

- W1 - There seems to be a serious disconnect between the theory proposed in the beginning of section 4 and the actual model used.  The theory marginalizes all future gazes; the model computes an anticipated future gaze.  The theory has no notion of HOI; the model heavily depends on it. ($\mathbf{y}$ are defined as activity labels not HOI predictions, so the description at L214-5 is unclear as "activity labels" are not sufficiently defined.)

- W2 - Similarly, it is not clear whether the model is completely trained together or if it is trained in parts.  Is all of the data used to train all of the parameters, even though the action parts are conditioned on the gaze parts?  Or is there some sequencing used to train the pieces of the model.  This explanation is missing.

- W3 - Lack of detailed analysis w.r.t. the value of the gaze anticipation in the work, incl. the ExoCook dataset.  A key question is how much knowing the future gaze leads to better action prediction.  There is no notion of upper bound computed, although it seems possible: the future gazes are known and could be incorporated instead of the module that predicts the gaze to measure how critical the gaze is for the action pieces.  I.e., what would performance be with perfect gaze?  Does it matter?  Is the improvement in actions idiosyncratic of other parts of the very big model.

- W4 - Similarly, there is no analysis of the anticipation modules.  These seem to have been engineered/learned to anticipate at a certain distance into the future (relatively short from what the manuscript indicates).  But, one wants to understand the robustness of these anticipation modules relative to the complexity of the dataset.  Half a second is very short.  In this time, one would want to understand the performance relative to the "null" baseline (i.e., no prediction, just copy the same gaze).

- W5 - Beyond the anticipation model analysis, the results are full of comparisons to the literature, but they neither sufficiently pull out certain pieces nor consider actual relevant baselines.  Perhaps there is a hope that the comparative papers covers these, but they leave multiple questions unanswered.  E.g., is HOI actually relevant here?  What if one just used raw pixels for the classification.  Where is the increase in performance really coming from (this is the key question)?

- W6 - The claim of generalization across viewpoints is discussed in 4.5 but suggests greater generality than what is evaluated.  FPV and TPV are separately modeled and evaluated.  Real adaptability would have one model applied to both tasks and able to adapt between them.  This is either overstated, or unclear.

Minor
- the formatting of the references in many parts of the paper is wrong. First type of mistake: "Li et al. Li et al. (2021)". Second type of mistake: L152 "The Ego-Exo4D data Grauman et al. (2024); either a "by" is needed or put the whole thing in parentheses.  This is distracting.
- Is it Exo-Cook or ExoCook?
- Fix the double-quotations (latex).
- "through a single, unified cognitive model in the brain" This is a claim, not a fact, and hence needs to be substantiated.  Actually most of the first paragraph is claims without substantiation.

**Questions:**

- What is the variability of the views and complexity of the actions in the snippets in ExoCook?

---

### Note · Authors · 2025-11-14

I have read and agree with the venue's withdrawal policy on behalf of myself and my co-authors.